**Data Availability Statement:** The informed consent document for this study does not allow for us to make the qualitative data (collected from a

# "It really proves to us that we are still valuable": Qualitative research to inform a safer conception intervention for men living with HIV in South Africa

**Lynn T. Matthews**[1,2]*, **Letitia Greener**[3,4], **Hazar Khidir**[5], **Christina Psaros**[6], **Abigail Harrison**[7], **F. Nzwakie Mosery**[3], **Mxolisi Mathenjwa**[3], **Kasey O'Neil**[2], **Cecilia Milford**[3], **Steven A. Safren**[8], **David R. Bangsberg**[9], **Jennifer A. Smit**[3]

**1** University of Alabama at Birmingham, Birmingham, AL, United States of America, **2** Center for Global Health and Division of Infectious Diseases, Massachusetts General Hospital, Boston, MA, United States of America, **3** Department of Obstetrics and Gynaecology, MRU (MatCH), University of the Witwatersrand, Faculty of Health Sciences, Durban, South Africa, **4** Wits Reproductive Health and HIV Institute (Wits RHI), Faculty of Health Sciences, University of the Witwatersrand, Johannesburg, South Africa, **5** Harvard Medical School, Massachusetts General Hospital, Brigham and Women's Hospital, Combined Residency Program in Emergency Medicine, Boston, MA, United States of America, **6** Department of Psychiatry, Massachusetts General Hospital, Behavioral Medicine, Boston, MA, United States of America, **7** Department of Behavioral and Social Sciences, Brown University School of Public Health, Providence, RI, United States of America, **8** Department of Psychology, University of Miami, Coral Gables, FL, United States of America, **9** Oregon Health & Science University-Portland State University School of Public Health, Portland, OR, United States of America

* lynnmatthews@uabmc.edu

## Abstract

### Objective

Many men living with HIV want to have children. Opportunities to reduce periconception HIV transmission include antiretroviral therapy as prevention, pre-exposure prophylaxis, limiting condomless sex to peak fertility, and sperm processing. Whether men have knowledge of or want to adopt these strategies remains unknown.

### Methods

We conducted focus group discussions (FGDs) with men accessing HIV care in South Africa in 2014 to inform a safer conception intervention for men. Eligible men were 25–45 years old, living with HIV, not yet accessing treatment, and wanting to have a child with an HIV-negative or unknown serostatus female partner (referred to as the "desired pregnancy partner"). FGDs explored motivations for having a healthy baby, feasibility of a clinic-based safer conception intervention, and acceptability of safer conception strategies. Data were analyzed using thematic analysis.

### Results

Twelve participants from three FGDs had a median age of 37 (range 23–45) years, reported a median of 2 (range 1–4) sexual partners, and 1 (range 1–3) desired pregnancy partner(s). A third (N = 4) had disclosed HIV-serostatus to the pregnancy partner. Emergent themes

small sample of men) publicly available. Data access requests for elements of raw data may be sent to the UAB Center for Clinical and Translational Science via CCTS@uab.edu; primary study authors may also be contacted.

Funding: This work was supported by National Institutes of Health (NIH, https://urldefense. proofpoint.com/v2/url?u=http-3A__www.nih. gov&d=DwIGaQ&c=o3PTkfaYAd6- No7SurnLt5qpge1aKYwPQyBFS7c8AA0&r= gogn30blU2aosnJjpi727b9K-qufLi2BKLsFiLaeo_ U&m=86VO_njWvYfHhG- 1S2zTCw2keBxdMZiKedjcrviJsKo&s= 4cB5P8jmZRi23kr3nk3zngI9MgCA2hQ_ fGdAPBnYJps&e=) awards R34 (MH100948) (LTM), K23 (MH095655) (LTM), K23 (MH096651) (CP), K24 (MH087227) (DRB), K24 (MH094214) (SS), as well as the Harvard CFAR P30 (AI060354) (LTM). This work does not necessarily reflect the opinions of the NIH. The funders had no role in study design, data collection and analysis, decision to publish, or preparation of the manuscript.

Competing interests: The authors have declared that no competing interests exist.

included opportunities for and challenges to engaging men in safer conception services. Opportunities included enthusiasm for a clinic-based safer conception intervention and acceptance of some safer conception strategies. Challenges included poor understanding of safer conception strategies, unfamiliarity with risk reduction [versus "safe" (condoms) and "unsafe" (condomless) sex], mixed acceptability of safer conception strategies, and concerns about disclosing HIV-serostatus to a partner.

## Conclusions

Men living with HIV expressed interest in safer conception and willingness to attend clinic programs. Imprecise prevention counseling messages make it difficult for men to conceptualize risk reduction. Effective safer conception programs should embrace clear language, e.g. undetectable = untransmittable (U = U), and support multiple approaches to serostatus disclosure to pregnancy partners.

## Introduction

In sub-Saharan Africa, up to 60% of new infections occur in stable, HIV-serodifferent partnerships, in which both partners often place great value on having children [1–7]. HIV-uninfected women who conceive with a partner living with HIV face HIV acquisition and perinatal transmission risks, highlighting the importance of periconception counseling for men living with HIV (MLWH) to prevent transmission [8–12]. Antiretroviral treatment (ART), pre-exposure prophylaxis (PrEP), limiting condomless sex to peak fertility, and sperm processing are effective HIV prevention strategies for HIV-serodifferent couples wherein the male partner is living with HIV and they want to have a child [13].

While MLWH control many reproductive decisions, they rarely receive safer conception advice [3, 14–19]. Studies show that more than half of men living with HIV entering the care system want children with their current partners, and men's reproductive goals are important aspects of their social, cultural, gender, and family dynamics [4, 11, 20–22]. Delivering safer conception messages to men addresses sexual power imbalances to promote HIV prevention and is consistent with a UNAIDS call for "a global shift in the discussion on HIV and gender— that it should become more inclusive of men. . .[23]."

Based on our formative work in KwaZulu-Natal, South Africa, we designed a "healthy baby" intervention to support MLWH to adopt HIV risk reduction behaviors to keep partners, and therefore their children, HIV-uninfected [24]. We conducted focus group discussions (FGDs) with MLWH wanting to have a child in the next year with a partner of unknown or HIV-negative serostatus in order to solicit their input into the healthy baby intervention. Here we describe insights into the feasibility and acceptability of a healthy baby or safer conception intervention for men. While these data were collected in 2014, men living with HIV remain a largely unreached population, their reproductive goals are not yet integrated into HIV care and we believe these data remain relevant to understanding the needs and considerations of men living with HIV in South Africa in 2020.

## Methods

### Description of study site

We recruited participants from an NGO/Department of Health (DoH) collaborative healthcare facility based in a large township in eThekwini, KwaZulu-Natal, South Africa.

## Selection of participants

HIV-positive men aged 20–45, with knowledge of their HIV-serostatus for at least 6 months, currently receiving care, not on ART, and interested in having a child in the next year with a stable partner of HIV-negative or unknown serostatus (referred to as the "desired pregnancy partner" in this manuscript) were eligible to participate. Participants were recruited and focus groups were conducted in July and August 2014. At that time, people living with HIV (PLWH) with CD4 count greater than 350 cells/mm3 were not eligible for treatment per South African DoH guidance [25]. Potential participants were recruited at a facility based in KwaZulu-Natal using convenience sampling and were screened based on eligibility criteria.

## Data collection

FGDs were employed to explore group insights and community norms regarding intervention content. Each FGD comprised 3–5 men, was conducted in isiZulu and audio-recorded. The FGD guides were semi-structured, informed by prior qualitative studies [15–17] and designed to explore: motivations for having a healthy baby, partnership dynamics and feasibility of engaging partners in a safer conception intervention, knowledge of and acceptability and feasibility of safer conception strategies available in the public sector, and the logistics of participating in a clinic-based safer conception intervention. The safer conception methods were described through images and a narrative (Fig 1) developed for a prior study and adapted to this setting through input from the study team, local clinic staff, and peer counselors [26]. The methods described included timing condomless sex to peak fertility, "early" ART (FGDs were conducted when national guidelines recommended treatment based on CD4 count <350 and WHO clinical stage), and PrEP (FGDs were conducted prior to local PrEP roll-out, after PrEP was recommended for persons at risk for HIV by the CDC and prior to recommendations for use by serodifferent couples from the WHO.) Sperm washing was not described because it is not available in the public sector in South Africa. However, it was discussed in each FGD session. A brief questionnaire captured socio-demographic information, sexual behavior, and fertility desires.

## Data analysis

FGD audio-recordings were transcribed and translated into English. The transcripts were independently read by three researchers who worked together with the focus group discussion leaders to develop a codebook. The coding was conducted by three researchers and analyzed using an iteratively-developed codebook to explore emergent themes using thematic analysis [27]. Emergent themes were summarized, then discussed and refined by the team and compared for consistency and discrepancies. Discussion of themes facilitated the identification of connections between the research questions, coding categories, and raw data. Quantitative data from the questionnaires are described using median (range) and number (%).

**Ethics statement.** Ethics approvals were obtained from the University of the Witwatersrand (Johannesburg, South Africa) and Partners Healthcare (Boston, MA). Healthcare facility support was obtained and participants provided written, voluntary informed consent.

## Results

Sixty-nine men accessing HIV care were approached, 18 were eligible, and 12 consented and participated. Principal reasons for ineligibility were HIV-serostatus known for less than six months and reporting a pregnancy partner living with HIV. Six eligible men who did not participate in FGDs had scheduling conflicts due to employment. Median age of enrolled

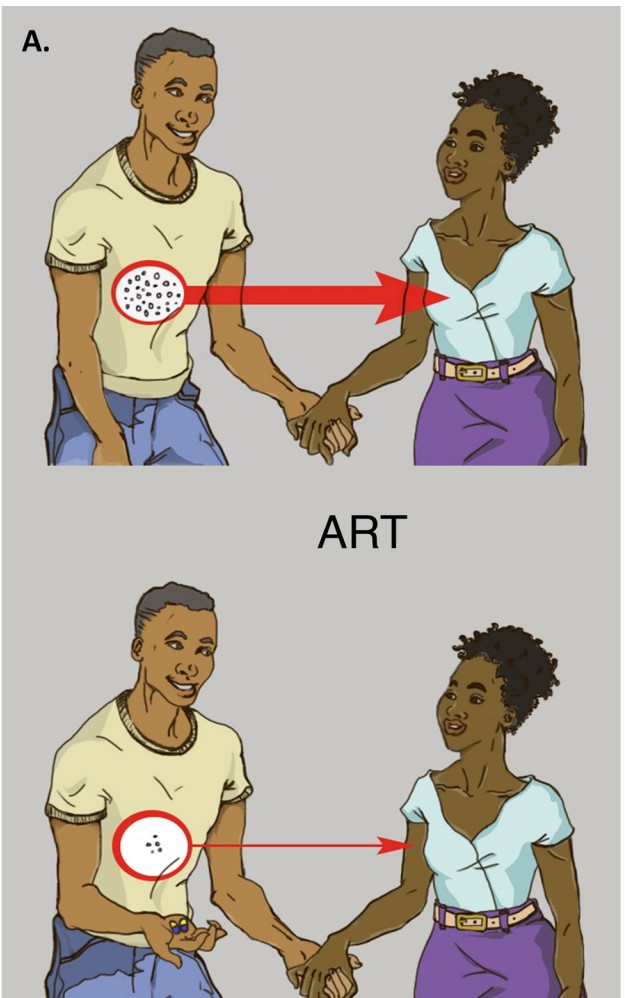
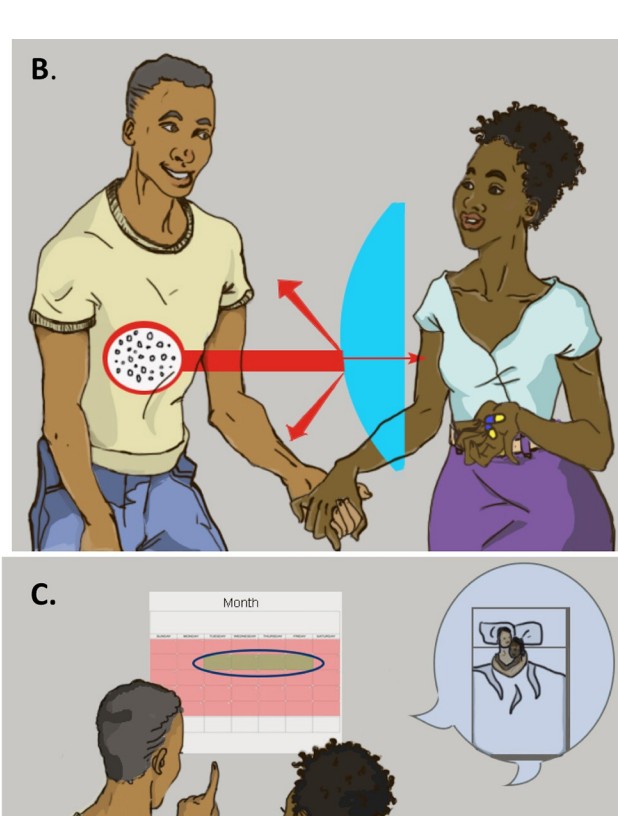
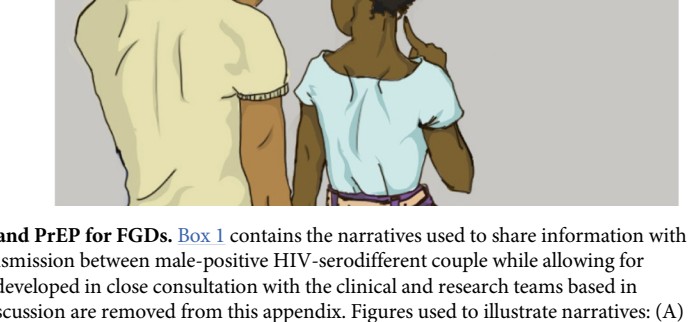

**Fig 1. Images and script describing timed condomless sex, anitretrovirals, and PrEP for FGDs.** Box 1 contains the narratives used to share information with participants about the various safer conception strategies to reduce sexual transmission between male-positive HIV-serodifferent couple while allowing for conception. Images accompanied each narrative. Images and narratives were developed in close consultation with the clinical and research teams based in South Africa. Questions embedded throughout the narratives to encourage discussion are removed from this appendix. Figures used to illustrate narratives: (A) ART as prevention, (B) PrEP, (C) Timing sex without condoms to peak fertility. Figures used to illustrate narratives (Reprinted under a CC BY license, with permission from Asha Pieper, original copyright 2015).

participants was 37 years, 75% were employed, and nearly half did not know the HIV-serostatus of their intended pregnancy partner (or the woman identified as the person with whom they hoped to have a child in the next year) (Table 1).

Emergent themes from the data are organized into opportunities and challenges that men identified for participating in a clinic-based safer conception program. Opportunities included enthusiasm for a safer conception program that would give men hope to continue living their lives and contribute to society as fathers and the acceptability of treatment as prevention and PrEP as safer conception strategies. Challenges included limited knowledge of safer conception strategies, poor understanding of "risk reduction", a desire to avoid long-term exposure to ART, and fear of serostatus disclosure to pregnancy partner(s) leading to relationship dissolution.

## Box 1. Narratives

I want to share with you some of the safer conception strategies that are available and ask your opinions about whether these could work for HIV-infected men here in [place name]. Let's say that Sipho is an HIV infected man and he wants to have a child with his partner Naledi. Naledi does not have HIV, and one of the ways to make sure Sipho has an uninfected baby is to protect Naledi from HIV.

Naledi is more likely to get pregnant during particular times in her menstrual cycle. One way to reduce the risk of transmitting HIV to Naledi while trying to get pregnant is to use condoms most of the time. But when Naledi is most fertile, the couple can have sex without condoms to try to conceive. This method is known as "**timed intercourse**" and can be used to reduce the risk of HIV transmission between partners, while still allowing the woman to become pregnant.

If Sipho is started on antiretrovirals (ARVs) and takes his pills every day, the level of HIV virus in his blood will be very low. When the HIV virus in his blood is low, it is unlikely that Sipho can transmit the virus to Naledi or to their baby. This is sometimes called **treatment as prevention**.

Another option, which is still being studied, would be for Naledi–who is uninfected–to take a kind of ARVs called **PrEP or pre-exposure prophylaxis**. With PrEP, an HIV-negative partner with an HIV-positive partner can take certain ARVs in order to reduce the chances of getting HIV. Doctors are still studying whether this method works and how PrEP might be used for couples who want to have a baby.

## A. Opportunities for safer conception services for men

1. An intervention to help men have a healthy baby allows men to live their lives and contribute to society. Men expressed enthusiasm for a safer conception program for men.

> **Participant 10 (P10):** "That is exactly what pushed me to come here today. . . ... If there is anything that I can use as an infected person, or anything that could protect my partner

**Table 1. Participant demographics.**

|  | N = 12 |
|---|---|
|  | **Median (min-max) Number (%)** |
| Age, years | 37 (23–45) |
| Employed | 9 (75%) |
| Number of children | 3 (0–6) |
| Identified one pregnancy partner | 11 (92%) |
| Length of relationship, years | 3 (1–13) |
| HIV-status of an intended pregnancy partner (N = 13) |  |
| Negative | 7 (54%) |
| Positive | 1 (8%) |
| Unknown | 5 (38%) |
| Disclosed HIV-serostatus to an intended pregnancy partner | 4 (31%) |
| Condom use at last sex (with 13 intended pregnancy partners) | 11 (85%) |

from getting infected if we engage in unprotected sex. Or perhaps something that I can use before I have sex with her. . . . We cannot have a baby if we use condoms." *-FGD3*

Few participants felt they could discuss reproductive goals with healthcare providers or their partners. Further, many had never heard of "safer conception"–or opportunities for conceiving a child with reduced HIV transmission risk to partner. Men noted that having a healthy baby would allow them to live, find hope, and be 'valuable' members of their community, family, and society. Their descriptions suggest the potential for safer conception care to ameliorate HIV-related stigma.

**P9**: "Today's discussion created an image . . . that shows that life goes on for men. And that there are still medical personnel who are trying that our partners do not get infected and that if we still want to have babies, there are methods that we need to follow so that we can have healthy babies even if we are in this situation.

**P10:** Being in this situation of being HIV positive, you end up losing hope. . . . So the presence of these programs and this research . . . Because it really proves to us that we are still valuable, there are things that are planned for us even though we are in this situation."– *Exchange from FGD3*

2. Treatment as prevention and PrEP are feasible safer conception strategies. When the moderator described safer conception strategies (Fig 1), men identified possible benefits to adopting these strategies.

A few men expressed that "early" ART initiation would be a feasible strategy to promote their health, protect their partner from HIV, and ultimately have an uninfected child.

**P7**: "I commend this method, because it covers both issues simultaneously. It protects her from getting infected, at the same time she can get a baby. [. . .]

**P5**: Yes I also commend this method of starting treatment earlier so that the viral load can be low and increase chances of not infecting your partner easily."–***Exchange from FGD2***

Some felt that partner support, including her knowledge of the man's HIV serostatus, would be a strong motivator to initiate and adhere to treatment by enhancing social support and decreasing the need for secrecy.

**P10:** "[S]he would know the reason behind me starting treatment early. If she is also positive about the reason for me to start treatment earlier, I do not think it would be a problem." *-FGD 3*

Some men were enthusiastic about the idea of PrEP as prevention for the partner, although the assumptions they make about partner enthusiasm may have limited validity.

**P3:** "I think my partner has to feel alright because this method helps her not to get the virus. [. . .] I do not think she can have a problem with it because I think she also does not want to be infected." *-FGD1*

In each of the FGDs, men found limiting condomless sex to peak fertility compelling but quickly realized that this strategy would not be feasible without serostatus disclosure.

**P5:** "Yes, it would work for someone who is brave [enough] to talk with his partner [and] disclose his status early. . .because she would want to know, 'Why are we not using a condom today after we have been using it all along? . . . So why don't we engage in unprotected sex all the time?,' then you have to explain."–**FGD2**

Men expressed that while these methods were mostly unfamiliar, they and their partners were trusting of information coming from within the healthcare system.

## B. Challenges for safer conception services for men

**3. Poor understanding of safer conception techniques and the concept of risk reduction.**  In each FGD, methods to prevent perinatal transmission from a mother living with HIV were raised by participants as "healthy baby" strategies.

**P5:** "I know that the mother of the baby consumes a pill when she is closer to giving birth, which will protect the baby from being infected."–**FGD2**

When men were encouraged to think of ways to protect a partner from HIV while conceiving a child, they described sperm washing, in vitro fertilization, and donor sperm as strategies they had heard of but did not know how to access.

**P10:** "Perhaps the little information that I have is that my sperm is taken and inserted into the female so that the woman can conceive. I am not sure what that is called. . . ."–**FGD3**

Additional strategies such as ART for the infected partner, PrEP for the uninfected partner, limiting condomless sex to peak fertility were not familiar. Men were unfamiliar with the concept of risk reduction, expressing that condom use was 100% effective and that methods to reduce transmission risk in the absence of condoms were unlikely to protect a partner from acquiring HIV. In this example, a participant expresses skepticism about ART-mediated HIV-RNA suppression.

**P2:** "It [the virus] is there but it is lowered, it never gets completely washed away. Eish chances of her not getting infected are very low man, they are very low because it is there."—**FGD1**

**4. Barriers to uptake of safer conception strategies: desire to avoid ARVs, doubts of efficacy.**  Men described strong desires to avoid antiretrovirals, as treatment for prevention. Starting ART was seen as contrary to what had been advised by providers (at the time of these interviews the guidelines did not recommend treatment for asymptomatic PLWH with CD4 cell counts above 350), was a 'forever' commitment, and associated with worse health. Men believed 'healthy living' could maintain a high CD4 count and suppress HIV RNA without initiating ART.

**P1:** "But the moment you start on treatment it means your health is not stable. [. . .] I am not prepared to start on treatment. [. . .] What I am prepared to do which is what I was told . . . to eat healthy food, exercise, that is exactly what makes your CD4-count to be high."–**FGD1**

**P9:** "Personally, I think if I could wait until the viral load decreases on its own without having used the pills. . . . because pills are not something that I like."–**FGD3**

Men had reservations about uptake of an investigational (at the time of these discussions) strategy such as PrEP. The idea of taking medication was again associated with ill health and some men expressed concerns that a healthy partner would not want to take a daily prophylactic medication.

**P7:** "I do not think she would agree, to take medication when she is not sick."–*FGD2*

Men also explained that partners might not understand that HIV-RNA suppression would keep them protected from HIV.

**P6:** "Women do not consider how high or low the viral load is in your body, if there is sign of being 'positive'. They only consider being positive or negative, they do not care about how high or low the viral load is."–*FGD2*

**5. Feasibility of disclosure in order to implement safer conception strategies: Challenges of HIV-serostatus disclosure to partner.** Men identified disclosure as a critical part of implementing safer conception strategies.

**P9:** "What I would mention as challenges would be . . . it is that a person would know his status while not knowing his partners' status, so he might be scared to attend such meetings, then his partner would ask what is about and what you are doing there. Because males, in most cases, they hide their status."–*FGD3*

Men described an environment that made it difficult to imagine disclosing serostatus and expressed lacking knowledge on how to approach disclosure.

**P5:** "I am afraid of telling my partner, I do not know how I can do that. I planned that I will never tell her".

**P7:** "It is difficult to disclose to your partner if you are positive and your partner is negative. You just do not know how you can start telling her that."–*FGD2*

Participants described challenges to disclosing due to fears of relationship dissolution and stigma.

**P2:** "There is that fear that you might lose her, so it is hard. . . . I won't lie, my sister. Even now I do not know how I can break the ice. I don't know, because I think if I tell her my status she will leave me. . . .. I am hesitant because I am scared." -*FGD1*

While most of participants were anxious about HIV-serostatus disclosure to a partner, they identified the healthcare system as a place to assist people living with HIV who need support to disclose:

**P4:** "Visiting clinics is what can help them. [. . .] They should go as a couple. They should not go one by one. Because even to us coming here alone, then telling our partners, is a big problem. It would be better if both are on the same page so that there could be an understanding."—*FGD2*

Despite barriers to disclosure identified in each discussion group, there were some positive examples of disclosure by men.

**P9:** "I couldn't keep it as a secret and I decided to tell my partner, even though it is not easy to tell people. And I told her, she accepted, and she said "okay" and she understood. Life goes on, even when it comes to what my fellow brothers were talking about makes sense and we are moving forward and I will be able to have a baby, . . . Speaking out helps."– *FGD3*

## Discussion

Men are eager for services to support their reproductive goals [17, 19, 21, 28]. Participants in this study, men living with HIV, engaged in care and not yet accessing ART, were eager to access safer conception services and maintained that other men in their communities would be interested. Mixed acceptability of the safer conception strategies suggests the importance of method choice [13, 20, 21]. Given that fewer than a third of men had disclosed their serostatus to their desired pregnancy partners and all FGDs identified numerous barriers to serostatus disclosure, our findings highlight the importance of and challenges to supporting HIV-serostatus disclosure and communication within safer conception care.

MLWH are less likely to engage and remain in care, suppress HIV-RNA, and survive compared to women [29–32]. The enthusiasm expressed by men suggests that offering men comprehensive reproductive healthcare may be a novel patient-centered strategy to help them engage in HIV care and treatment services, suppress HIV-RNA and live longer, healthier lives. Ongoing calls for promoting equitable care for men who have sex with women highlight the need to create care that supports the needs of men as well as women in order to meet important 90-90-90 and 95-95-95 targets [33, 34]. Providing services to men may promote disclosure and also engage HIV-exposed women into HIV prevention services. This is supported by the first demonstration project of safer conception services in South Africa wherein all men (N = 192) participated with their pregnancy partners in a counseling intervention based in Johanneburg [35, 36]. However, recruiting men to attend FGD sessions at a fixed time was challenging given commitments, mostly related to keeping or seeking employment and highlight one of the challenges to engaging men in a facility-based program with traditional hours [37, 38].

Men have low knowledge of how to reduce periconception HIV transmission. In spite of safer conception guidelines in South Africa, providers rarely counsel people living with HIV, especially men [15, 16, 39, 40]. PLWH do not ask about options for pregnancy for many reasons, including sensing that providers are not supportive of their reproductive goals and stigma towards PLWH having children [4, 16, 40]. The basic tenets of safer conception (e.g. delay conception attempts until HIV-RNA is suppressed, PrEP for uninfected partners) should be offered to all PLWH rather than limited to those who identify as planning to have a child [13]. This aligns with broader treatment as prevention goals in South Africa and globally to promote viral suppression for all people living with HIV as a key strategy to improve health and prevent ongoing transmission.

When we described the motivation for a "healthy baby" intervention and the different safer conception methods, men found the principles of safer conception compelling, with mixed acceptability. This is consistent with data from other studies in South Africa [20, 41] and illustrates the importance of method choice (e.g. ART for the infected partner may not be the right choice for everyone) [42]. A safer conception intervention will likely require several sessions given the information gaps and may require creative approaches to education [43]. Implementation programs are needed to assess the methods which individuals and couples are likely to use and what support they will need to do so successfully [42, 44]. In addition, since these

focus groups were conducted, the data that undetectable is untransmittable (U = U) have grown substantially [45, 46]. While our data precede U = U messaging and universal test and treat policy in South Africa, surveys suggest a lag in communicating these messages to people living with and at-risk for acquiring HIV [47, 48] and leading advocates and health services researchers urge for clear language to convey the zero transmission risk associated with viral suppression [49]. Clearer, patient-centered messages would address the challenges that men described in understanding more conservative language about risk reduction. In addition, Universal Test and Treat (UTT) strategies highlight ongoing challenges of HIV care for men [50–54]. In South Africa, UTT implementation was followed by increases in women accessing ART, leading to HIV incidence declines for men while women remain at high risk for infection [55–57] suggesting that the lessons we can learn from this work remain relevant. Recent calls for gender equity that includes men more purposively in healthcare in South Africa highlight the importance of male-friendly services [33, 34].

While fewer than a third of participants had disclosed their HIV status to their pregnancy partner, through the discussions men concluded that disclosure would be an important part of safer conception while acknowledging the challenges with this important step. Mutual disclosure depends on both partners undergoing HIV counseling and testing (HCT) and sharing their results or completing couples-based counselling and testing (CHCT) together. HCT uptake in South Africa remains low, especially among men [58–60]. CHCT is not widely available in South Africa [61]. CHCT and other strategies to test partners (e.g. self screening [62]) are critical to promote uptake of ART for partners living with HIV and PrEP for those who remain uninfected with ongoing HIV exposure. Framing disclosure and CHCT within a holistic approach that focuses on building healthy families may support couples to overcome barriers to disclosure, and testing of this hypothesis is ongoing [24, 63]. In the meantime, it is important to offer safer conception counseling, education and support to individuals as well as mutually-disclosed couples [64, 65].

Men expressed trust in the healthcare system and anticipated that their partners would also trust information delivered in the clinic. Men simultaneously expressed difficulty navigating discussions about reproductive goals with providers; highlighting the need for healthcare staff who have skills to support MLWH to explore and initiate safer conception strategies [40]. A client-centered, peer-based approach that allows for the flexibility of one-on-one as well as couples based interactions may enhance support and reduce potential anxieties around new methods may provide support, empathy and affirmation of reproductive goals among MLWH. Many of the barriers that men identified to implementing safer conception strategies were wrought with anxiety (e.g. would HIV RNA suppression really protect a partner in the setting of condomless sex?) and fear (e.g. a lifelong commitment to ART, disclosure to partner). In order to overcome these barriers, a cognitive behavioral approach that provides information, assesses motivation, and employs problem-solving to address barriers to implementing safer conception strategies may be an effective means of supporting men to consider, adopt, and adhere to safer conception strategies [66].

The limitations of this study generalize to qualitative research–a small sample size means results are meant to generate hypotheses for future research. The discussions were conducted prior to implementation of test and treat policies–so negative attitudes towards treatment as prevention may have since lessened. We had particular challenges recruiting men for this study–highlighting the challenges of engaging men in clinic-based studies. The discussion groups were small, but the men shared rich data which aligns with findings from other work [17, 24, 67]. Smaller groups may have promoted more open discussion about a controversial topic [68]. Men who participated in our FGDs participated in 2014 and had a particular interest in this topic and may not reflect the views of all men in the community. However, given the

enthusiasm for engaging men in HIV care and reproductive health and the paucity of data about how to do so, these limited data remain important to disseminate to inform future studies [23, 69, 70].

These data suggest that men are eager to engage in safer conception care. Integrating effective combination HIV prevention into comprehensive reproductive health programs for men and women living with HIV provides an opportunity to reduce periconception HIV-transmission risk, support the reproductive rights of men and women living with and affected by HIV, while promoting the health of men, their partners, and their families [13].

## Supporting information

**S1 File. Safer conception men FGD guide.**
(PDF)

## Author Contributions

**Conceptualization:** Lynn T. Matthews, Letitia Greener, Christina Psaros, Abigail Harrison, Cecilia Milford, Steven A. Safren, David R. Bangsberg, Jennifer A. Smit.

**Data curation:** Lynn T. Matthews, Letitia Greener, Christina Psaros, Abigail Harrison, F. Nzwakie Mosery, Mxolisi Mathenjwa, Cecilia Milford, Steven A. Safren, David R. Bangsberg, Jennifer A. Smit.

**Formal analysis:** Lynn T. Matthews, Hazar Khidir, Christina Psaros, Abigail Harrison, F. Nzwakie Mosery, Mxolisi Mathenjwa, Kasey O'Neil, Cecilia Milford, Steven A. Safren, David R. Bangsberg, Jennifer A. Smit.

**Funding acquisition:** Lynn T. Matthews.

**Investigation:** Lynn T. Matthews.

**Methodology:** Lynn T. Matthews.

**Project administration:** Lynn T. Matthews.

**Resources:** Lynn T. Matthews.

**Supervision:** Lynn T. Matthews.

**Writing – original draft:** Lynn T. Matthews, Letitia Greener, Hazar Khidir, Christina Psaros, Abigail Harrison, Kasey O'Neil, Cecilia Milford, Steven A. Safren, David R. Bangsberg, Jennifer A. Smit.

**Writing – review & editing:** Lynn T. Matthews, Letitia Greener, Hazar Khidir, Christina Psaros, Abigail Harrison, F. Nzwakie Mosery, Mxolisi Mathenjwa, Kasey O'Neil, Cecilia Milford, Steven A. Safren, David R. Bangsberg, Jennifer A. Smit.

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
