## [Decision Letter · Decision Letter 0]

24 Jun 2020

PONE-D-20-04292

“It really proves to us that we are still valuable”:  Qualitative research to inform a safer conception intervention for men living with HIV in South Africa

PLOS ONE

Dear Dr. Matthews,

Thank you for submitting your manuscript to PLOS ONE. After careful consideration, we feel that it has merit but does not fully meet PLOS ONE’s publication criteria as it currently stands. Therefore, we invite you to submit a revised version of the manuscript that addresses the points raised during the review process.

Although we are proceeding with one review, please note that the reviewer has raised some significant concerns regarding the methodological rigor of both the data collection and the analysis, as well as the relevance of the findings to the current policy and practice environment in South Africa. Please do carefully consider all of these issues in your revision.

We look forward to receiving your revised manuscript.

Kind regards,

Kristin Dunkle

Academic Editor

PLOS ONE

Journal Requirements:

2. We note that Figure 1 in your submission contain copyrighted images. All PLOS content is published under the Creative Commons Attribution License (CC BY 4.0), which means that the manuscript, images, and Supporting Information files will be freely available online, and any third party is permitted to access, download, copy, distribute, and use these materials in any way, even commercially, with proper attribution. For more information, see our copyright guidelines: http://journals.plos.org/plosone/s/licenses-and-copyright.

2.1.         You may seek permission from the original copyright holder of Figure 1 to publish the content specifically under the CC BY 4.0 license.

2.2.    If you are unable to obtain permission from the original copyright holder to publish these figures under the CC BY 4.0 license or if the copyright holder’s requirements are incompatible with the CC BY 4.0 license, please either i) remove the figure or ii) supply a replacement figure that complies with the CC BY 4.0 license. Please check copyright information on all replacement figures and update the figure caption with source information. If applicable, please specify in the figure caption text when a figure is similar but not identical to the original image and is therefore for illustrative purposes only.

3. Please include copies of the focus group discussion guide(s) used in the study, in both the original language and English, as Supporting Information, or include a citation if they have been published previously.

4.In your Data Availability statement, you have not specified where the minimal data set underlying the results described in your manuscript can be found. PLOS defines a study's minimal data set as the underlying data used to reach the conclusions drawn in the manuscript and any additional data required to replicate the reported study findings in their entirety. All PLOS journals require that the minimal data set be made fully available. For more information about our data policy, please see http://journals.plos.org/plosone/s/data-availability.

Reviewers' comments:

Reviewer's Responses to Questions

**Comments to the Author**

1. Is the manuscript technically sound, and do the data support the conclusions?

Reviewer #1: Partly

2. Has the statistical analysis been performed appropriately and rigorously? 

Reviewer #1: N/A

3. Have the authors made all data underlying the findings in their manuscript fully available?

Reviewer #1: No

4. Is the manuscript presented in an intelligible fashion and written in standard English?

Reviewer #1: Yes

5. Review Comments to the Author

Reviewer #1: The authors conducted a qualitative study using FGDs to gather data to use to inform a safer conception intervention for men living with HIV. This is an important area of research as there is urgent need for evidence-based strategies for linking and retaining men in HIV care in South Africa.

The manuscript is well-written and reads very well! I commend the authors for producing such a well-written paper. However, I have a couple of comments and suggestions that I hope the authors will consider in strengthening this paper.

METHODS

Page 6, line 37: Authors describe that 'Each FGD comprised 3-5 men, was conducted in isiZulu and

audio-recorded'. The number of men participating in each FGD were very small. I would like the authors to justify the acceptability of this in qualitative research as ideally the number of participants in a FGD should be higher than this. Furthermore, could the authors comment on whether this limited or strengthened the quality and or richness of the data gathered in this study?

RESULTS

While the findings are well presented, my view is that they were particularly descriptive, and lacking the depth that one would expect from a qualitative paper in which data was analysed using thematic analysis. Could the authors comment on this?

From reading the findings of this paper, some of the data presented in the manuscript support my view that the findings of this study are outdated and not relevant to where the HIV field is now in South Africa, especially with regards to UTT. For example, the two extracts presented on p14 line 168 and line 172 clearly point to the time when UTT had not been introduced in South Africa. While the authors have mentioned this in the limitations section of the paper, I would like the authors to provide a strong justification of the relevance and significance of these findings and their contribution to knowledge. Said in another way, could the authors highlight how this paper, if published, would advance knowledge on how to link and retain men in HIV care in South Africa and similar settings.

Thank you.

6. PLOS authors have the option to publish the peer review history of their article (what does this mean?). If published, this will include your full peer review and any attached files.

Reviewer #1: Yes: Yandisa Sikweyiya

---

## [Author Response · Author response to Decision Letter 0]

14 Aug 2020

The information below is easier to read in teh uploaded response to review due to formatting. However, the response is also here.

2. We note that Figure 1 in your submission contain copyrighted images. All PLOS content is published under the Creative Commons Attribution License (CC BY 4.0), which means that the manuscript, images, and Supporting Information files will be freely available online, and any third party is permitted to access, download, copy, distribute, and use these materials in any way, even commercially, with proper attribution. For more information, see our copyright guidelines: http://journals.plos.org/plosone/s/licenses-and-copyright.

2.1. You may seek permission from the original copyright holder of Figure 1 to publish the content specifically under the CC BY 4.0 license.

2.2. If you are unable to obtain permission from the original copyright holder to publish these figures under the CC BY 4.0 license or if the copyright holder’s requirements are incompatible with the CC BY 4.0 license, please either i) remove the figure or ii) supply a replacement figure that complies with the CC BY 4.0 license. Please check copyright information on all replacement figures and update the figure caption with source information. If applicable, please specify in the figure caption text when a figure is similar but not identical to the original image and is therefore for illustrative purposes only. 

Response. We have uploaded the form described in 1.1. We have added the text for Figure 1, including copyright year.

3. Please include copies of the focus group discussion guide(s) used in the study, in both the original language and English, as Supporting Information, or include a citation if they have been published previously.

Response. These are now included and referred to as supporting information in the text.

4.In your Data Availability statement, you have not specified where the minimal data set underlying the results described in your manuscript can be found. PLOS defines a study's minimal data set as the underlying data used to reach the conclusions drawn in the manuscript and any additional data required to replicate the reported study findings in their entirety. All PLOS journals require that the minimal data set be made fully available. For more information about our data policy, please see http://journals.plos.org/plosone/s/data-availability.

Response. These data were collected prior to the movement towards public sharing of qualitative data. The informed consent signed by men who participated in this study noted (bolding added for current purposes and not in signed ICF):

“You will be asked to allow the facilitator to digitally record the focus group discussion, so that the study staff can make sure that it is being carried out correctly and that they understand what is being said by participants. Each digital recording will be transcribed, and all recordings will be erased within two years of publication of study findings, or if there is no publication, no later than six years after the study has ended. Information from the recordings/disks may be presented at professional meetings or in written articles, in which case no names or other personal identifiers will be used. 

Digital recording is a requirement for study participation. The focus group discussion will be confidential; it will be identified only by a unique number assigned to you, and no individual names will appear on the audio file or the transcript of the focus group discussion. No one, except the researchers, will have access to the audio file or the transcript of the focus group discussion. You can decide to withdraw from the focus group discussion at any time. If you do not want the focus group discussion to be digitally recorded, you are not eligible to participate in the research study.”

Based on this ICF we do not have permission from the participants to make the full transcripts publicly accessible. We are unable to reach the men to ask for additional permissions at this time. We would be able to carefully redact and share portions of the dataset (not full transcripts given the signed consent documentation) with interested researchers with specific requests to the study authors. 

5. Review Comments to the Author

Reviewer #1: The authors conducted a qualitative study using FGDs to gather data to use to inform a safer conception intervention for men living with HIV. This is an important area of research as there is urgent need for evidence-based strategies for linking and retaining men in HIV care in South Africa.

The manuscript is well-written and reads very well! I commend the authors for producing such a well-written paper. However, I have a couple of comments and suggestions that I hope the authors will consider in strengthening this paper.

METHODS

Page 6, line 37: Authors describe that 'Each FGD comprised 3-5 men, was conducted in isiZulu and

audio-recorded'. The number of men participating in each FGD were very small. I would like the authors to justify the acceptability of this in qualitative research as ideally the number of participants in a FGD should be higher than this. Furthermore, could the authors comment on whether this limited or strengthened the quality and or richness of the data gathered in this study?

Response. Thank you for this comment. We aimed to conduct focus group discussions with larger groups of men, however attendance was limited. We believe that the small numbers highlight the challenges of conducting research with this population and while the small sample size is a weakness, the men who did attend were engaged and shared rich data. In addition, their comments were aligned with what men living with HIV communicate about these topics in our other datasets and thus we do not think the size of the focus groups limited the relevance of the information collected. We have added more discussion re. the sample size to the discussion.

RESULTS

While the findings are well presented, my view is that they were particularly descriptive, and lacking the depth that one would expect from a qualitative paper in which data was analysed using thematic analysis. Could the authors comment on this?

Response. We agree. The focus groups were designed to solicit feedback on the intervention we were designing. FGDs explored motivations for having a healthy baby, feasibility of a clinic-based safer conception intervention, and acceptability of safer conception strategies. We had already conducted open ended in depth interviews with men affected by HIV in prior research. Therefore the questions in this discussion guide were slightly more focused and as such the analysis is a bit more descriptive. However, given this important comment, we have re-organized the analysis somewhat so that some of the more thematic elements of the analysis are clearer in the Results. 

From reading the findings of this paper, some of the data presented in the manuscript support my view that the findings of this study are outdated and not relevant to where the HIV field is now in South Africa, especially with regards to UTT. For example, the two extracts presented on p14 line 168 and line 172 clearly point to the time when UTT had not been introduced in South Africa. While the authors have mentioned this in the limitations section of the paper, I would like the authors to provide a strong justification of the relevance and significance of these findings and their contribution to knowledge. Said in another way, could the authors highlight how this paper, if published, would advance knowledge on how to link and retain men in HIV care in South Africa and similar settings.

Response. We have added additional text highlighting that in the current era of UTT and TASP there remain glaring gaps in HIV care engagement and retention for men. There are no data in the literature to suggest progress in terms of men’s ability to disclose to partners. A recent (published while working on this response) JIAS supplement focused entirely on the gaps of engaging men who have sex with women in care and largely on the South African setting – we have cited some of this work. We believe our findings continue to speak to the need for patient-centered care for men in South Africa and the possibility for addressing men’s reproductive goals to address this HIV care and treatment and partners serostatus disclosure for men.

---

## [Decision Letter · Decision Letter 1]

24 Sep 2020

PONE-D-20-04292R1

“It really proves to us that we are still valuable”:  Qualitative research to inform a safer conception intervention for men living with HIV in South Africa

PLOS ONE

Dear Dr. Matthews,

Thank you for submitting your manuscript to PLOS ONE. After careful consideration, we feel that it has merit but does not quite meet PLOS ONE’s publication criteria as it currently stands. Therefore, we invite you to submit a revised version of the manuscript that addresses minor points raised during the review process. Please see the additional and helpful feedback from the new reivewer.

We look forward to receiving your revised manuscript.

Kind regards,

Kristin Dunkle

Academic Editor

PLOS ONE

Reviewers' comments:

Reviewer's Responses to Questions

**Comments to the Author**

1. If the authors have adequately addressed your comments raised in a previous round of review and you feel that this manuscript is now acceptable for publication, you may indicate that here to bypass the “Comments to the Author” section, enter your conflict of interest statement in the “Confidential to Editor” section, and submit your "Accept" recommendation.

Reviewer #1: (No Response)

Reviewer #2: (No Response)

2. Is the manuscript technically sound, and do the data support the conclusions?

Reviewer #1: (No Response)

Reviewer #2: Yes

3. Has the statistical analysis been performed appropriately and rigorously? 

Reviewer #1: (No Response)

Reviewer #2: Yes

4. Have the authors made all data underlying the findings in their manuscript fully available?

Reviewer #1: (No Response)

Reviewer #2: Yes

5. Is the manuscript presented in an intelligible fashion and written in standard English?

Reviewer #1: (No Response)

Reviewer #2: Yes

6. Review Comments to the Author

Reviewer #1: I wish to thank the authors for the excellent job that they have done in revising the manuscript. It has now become stronger especially the Results and Discussion sections.

I was pleased to note that the study limitations including those of the data used in this paper have now been expanded and discussed in the Discussion section.

My comments I raised in the previous round of reviews have been adequately addressed.

Thank you.

Reviewer #2: Thank you for your paper it is a very important topic which is not widely written about, and the paper is very well written. Nonetheless, I offer a few thoughts below which I hope resonate and may enrich the paper further.

I do have a small concern that the data is from 2014, HIV knowledge and treatment access has shifted significantly in this time – how might this shift outcomes? I see that the authors address this in the discussion, however as the reader until I get to the discussion this issue was ‘shouting very loudly at me’. Perhaps you could briefly acknowledge this earlier on and then discuss it in detail in the discussions section?

Abstract: I have a few points here, please apply appropriately throughout the article as well.

“Twelve participants from three FGDs had a median age of 37 (range 23-45) years, reported a median of 2 (range 1-4) sexual partners, and 1 (range 1-3) desired pregnancy partner(s). A third (N=4) had disclosed HIV-serostatus to the pregnancy partner. “ this is a bit confusing, what does ‘desired pregnancy partners’ mean? And please explain a ‘pregnancy partner’. This comes up in Table 1 as well, please clarify the term, I assume it means they have a partner who they would want to conceive with?

“Conclusions: Men living with HIV are interested in safer conception and willing to attend clinic programs.“ please think about whether you can generalize to ‘Men living with HIV…” as you have done here – qualitative research provides incredibly important information, however, there are many reasons why we tend not to generalize in this way. Perhaps re-phrase to “These men living with…”

Selection of partners: I assume they were receiving care in the public health sector? Please specify.

“Potential participants were recruited at a facility 69 based in KwaZulu-Natal” – again I assume a public facility? Please specify.

Data analysis: “Quantitative data from the questionnaires are described.” please explain in more detail.

Results: the authors note that “When men were encouraged to think of ways to protect a partner from HIV while conceiving they described sperm washing, in vitro fertilization, and donor sperm as strategies they had heard of but did not know how to access.” It is very interesting, and quite surprising, that they had heard of these methods, but not the other methods – could you perhaps indicate how many had heard of it, to give a fuller picture, and perhaps also a comment about how much they knew and how accurate their understanding was.

Discussion: in your first paragraph you include the challenge of recruiting men and limited hours of facilities etc. While this is an important point, I don’t think it should be in this first paragraph, in my view it is not the most important point and should not be at the start of your discussion. I would shift it elsewhere.

You mention that “This is supported by the first demonstration project of safer conception services in South Africa wherein all men were able to engage their partners”, please specify when this was conducted and how many men were involved, to provide more context and understanding for the reader.

“PLWH do not ask about options for pregnancy for many reasons, including sensing that providers are not supportive of their reproductive goals” – can you expand on this? Is this stigma around PLWH or what?

Thank you this was an interesting read.

7. PLOS authors have the option to publish the peer review history of their article (what does this mean?). If published, this will include your full peer review and any attached files.

Reviewer #1: **Yes: **Yandisa Sikweyiya

Reviewer #2: No

---

## [Author Response · Author response to Decision Letter 1]

28 Sep 2020

Reviewer #1: I wish to thank the authors for the excellent job that they have done in revising the manuscript. It has now become stronger especially the Results and Discussion sections. I was pleased to note that the study limitations including those of the data used in this paper have now been expanded and discussed in the Discussion section. My comments I raised in the previous round of reviews have been adequately addressed. Thank you.

Response. Thank you for the helpful comments and for reviewing the revision.

Reviewer #2: Thank you for your paper it is a very important topic which is not widely written about, and the paper is very well written. Nonetheless, I offer a few thoughts below which I hope resonate and may enrich the paper further. I do have a small concern that the data is from 2014, HIV knowledge and treatment access has shifted significantly in this time – how might this shift outcomes? I see that the authors address this in the discussion, however as the reader until I get to the discussion this issue was ‘shouting very loudly at me’. Perhaps you could briefly acknowledge this earlier on and then discuss it in detail in the discussions section?

Response. Thank you for this comment. We added the following to the introduction: “While these data were collected in 2014, men living with HIV in South Africa remain an under-reached population, their reproductive goals are not yet integrated into HIV care, and we believe these data remain relevant to understanding the needs and considerations of men living with HIV in South Africa in 2020.”

Abstract: I have a few points here, please apply appropriately throughout the article as well.

“Twelve participants from three FGDs had a median age of 37 (range 23-45) years, reported a median of 2 (range 1-4) sexual partners, and 1 (range 1-3) desired pregnancy partner(s). A third (N=4) had disclosed HIV-serostatus to the pregnancy partner. “ this is a bit confusing, what does ‘desired pregnancy partners’ mean? And please explain a ‘pregnancy partner’. This comes up in Table 1 as well, please clarify the term, I assume it means they have a partner who they would want to conceive with?

Response. This study enrolled men planning for a pregnancy with a female partner in the next year. Therefore we asked questions about the enrolled men’s desired pregnancy partner. We added additional clarification of this term where first used in the methods and the abstract.

“Conclusions: Men living with HIV are interested in safer conception and willing to attend clinic programs.“ please think about whether you can generalize to ‘Men living with HIV…” as you have done here – qualitative research provides incredibly important information, however, there are many reasons why we tend not to generalize in this way. Perhaps re-phrase to “These men living with…”

Response. - We have edited this to ensure that readers understand that abstract conclusions refer to this data in the manuscript. This sentence now reads, “Men living with HIV expressed interest in safer conception and willingness to attend clinic programs…”.

Selection of partners: I assume they were receiving care in the public health sector? Please specify. Also, “Potential participants were recruited at a facility 69 based in KwaZulu-Natal” – again I assume a public facility? Please specify.

Response. In the methods section we indicated that men were recruited from an NGO/Department of Health (DoH) collaborative healthcare facility based in a large township in eThekwini, KwaZulu-Natal, South Africa. We did not recruit the partners of the men for the focus group discussions.

Data analysis: “Quantitative data from the questionnaires are described.” please explain in more detail.

Response. We have amended this sentence which now reads:

“Quantitative data from the questionnaires are described using median (range) and number (%).”

Results: the authors note that “When men were encouraged to think of ways to protect a partner from HIV while conceiving they described sperm washing, in vitro fertilization, and donor sperm as strategies they had heard of but did not know how to access.” It is very interesting, and quite surprising, that they had heard of these methods, but not the other methods – could you perhaps indicate how many had heard of it, to give a fuller picture, and perhaps also a comment about how much they knew and how accurate their understanding was.

Response. We added additional language to indicate what men shared about sperm washing. Because this is a qualitative paper with few men and we, therefore, did not systematically assess what all men knew about sperm washing (or any of the methods), we would prefer not to list the number of men who appeared to understand sperm washing as this is not really well aligned with the methods used in this manuscript.

Discussion: in your first paragraph you include the challenge of recruiting men and limited hours of facilities etc. While this is an important point, I don’t think it should be in this first paragraph, in my view it is not the most important point and should not be at the start of your discussion. I would shift it elsewhere.

Response. Thank you for this suggestion. We have moved this discussion to a later section on the challenges of engaging men.

You mention that “This is supported by the first demonstration project of safer conception services in South Africa wherein all men were able to engage their partners”, please specify when this was conducted and how many men were involved, to provide more context and understanding for the reader.

Response. Thank you for this suggestion. We have added more details about this project.

“PLWH do not ask about options for pregnancy for many reasons, including sensing that providers are not supportive of their reproductive goals” – can you expand on this? Is this stigma around PLWH or what?

Response. We have edited this sentence which now reads:

“PLWH do not ask about options for pregnancy for many reasons, including sensing that providers are not supportive of their reproductive goals and stigma towards PLWH having children (4, 16, 40).” 

Thank you this was an interesting read.

Response. Thank you for your helpful review!

---

## [Editor Report · Decision Letter 2]

7 Oct 2020

“It really proves to us that we are still valuable”:  Qualitative research to inform a safer conception intervention for men living with HIV in South Africa

PONE-D-20-04292R2

Dear Dr. Matthews,

We’re pleased to inform you that your manuscript has been judged scientifically suitable for publication and will be formally accepted for publication once it meets all outstanding technical requirements.

Kind regards,

Kristin Dunkle

Academic Editor

PLOS ONE
---

## [Editor Report · Acceptance letter]

18 Feb 2021

PONE-D-20-04292R2 

“It really proves to us that we are still valuable”:  Qualitative research to inform a safer conception intervention for men living with HIV in South Africa 

Dear Dr. Matthews:

I'm pleased to inform you that your manuscript has been deemed suitable for publication in PLOS ONE. Congratulations! Your manuscript is now with our production department. 

Kind regards, 

on behalf of

Dr. Kristin Dunkle 

Academic Editor

PLOS ONE